# OBJECT 3DIT:
# Language-guided 3D-aware Image Editing

**Oscar Michel**[1]    **Anand Bhattad**[2]    **Eli VanderBilt**[1]
**Ranjay Krishna**[1,3]    **Aniruddha Kembhavi**[1]    **Tanmay Gupta**[1]

[1]Allen Institute for Artificial Intelligence, [2]University of Illinois Urbana-Champaign,
[3]University of Washington

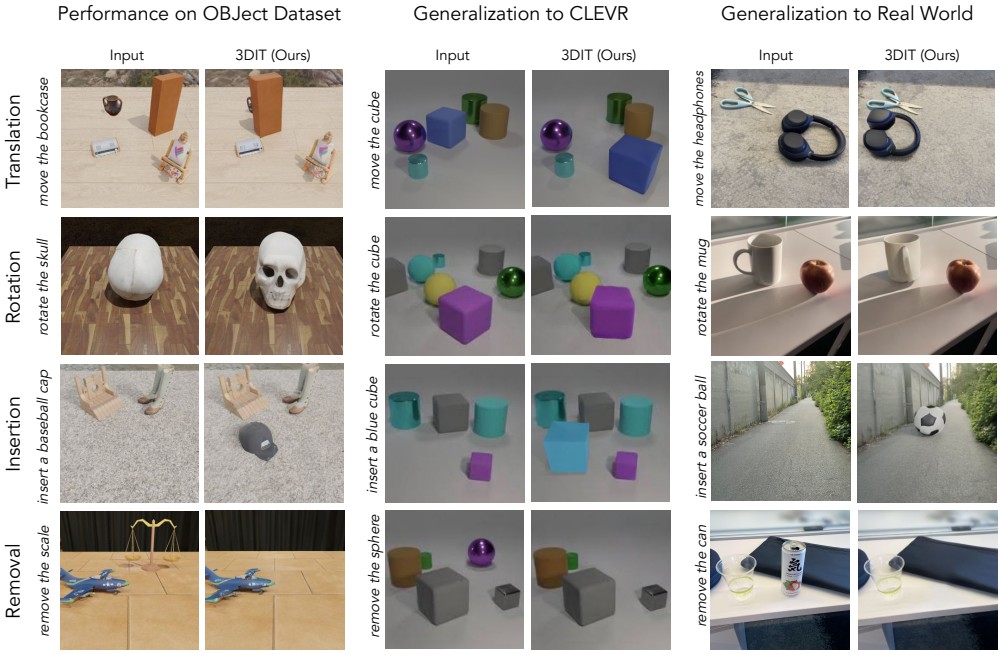

Figure 1: We present 3DIT, a model to edit individual objects in the context of a rich scene with language conditioning. 3DIT is able to effectively edit objects while considering their scale and viewpoint, is able to add, remove and edit shadows to be consistent with the scene lighting and is able to account for object occlusions. Training on our new benchmark OBJECT, 3DIT remarkably generalizes to images in the CLEVR dataset as well as the real world.

## Abstract

Existing image editing tools, while powerful, typically disregard the underlying 3D geometry from which the image is projected. As a result, edits made using these tools may become detached from the geometry and lighting conditions that are at the foundation of the image formation process. In this work, we formulate the new task of language-guided 3D-aware editing, where objects in an image should be edited according to a language instruction *in context* of the underlying 3D scene. To promote progress towards this goal, we release OBJECT: a dataset consisting of 400K editing examples created from procedurally generated 3D scenes. Each example consists of an input image, editing instruction in language, and the edited image. We also introduce 3DIT: single and multi-task models for four editing tasks. Our models show impressive abilities to understand the 3D composition

of entire scenes, factoring in surrounding objects, surfaces, lighting conditions, shadows, and physically-plausible object configurations. Surprisingly, training on only synthetic scenes from OBJECT, editing capabilities of 3DIT generalize to real-world images. More information can be found on the project page at https://prior.allenai.org/projects/object-edit.

# 1 Introduction

In today's visually-oriented society, the art of image editing has become an indispensable necessity. With the proliferation of camera phones and influences from social media platforms, amateur photographers want to transform ordinary snapshots into visual masterpieces. Unfortunately, the process of image editing is still in its infancy. Professional tools such as Photoshop allow pixel-level edits that can adjust lighting, insert objects, remove clutter, and introduce new shadows; however, these tools, with their steep learning curves are often daunting for novices. With the hopes of pulling image editors out from the minutiae of painstaking pixel-level edits, generative models have been heralded as a promise for object-level edits [57, 58, 31, 52].

Unfortunately, object-centric editing—translating or rotating an object while preserving the 3D geometry of the original photograph—is out of reach for generative models [29, 25, 73, 47, 70]. Although recent strides can take a segmented object and rotate and translate it, they typically operate on objects in isolation and often disregard any scene and lighting context [47, 54, 70]. Others require multiple viewpoints to reconstruct an object in 3D [6, 25, 73]. There is a need for models that can edit objects from a single image while preserving the structure of 3D objects and re-render shadows for the edited scene with the original lighting conditions.

To enable 3D-aware editing of objects in an image, we introduce OBJECT, **Obj**averse **E**diting in **C**ontex**T**, a large-scale benchmark to train and evaluate language-conditioned models that edit objects in images. We develop OBJECT by combining Objaverse [14], a recent 3D asset library, and Blender [12], a 3D rendering engine. OBJECT contains 400k editing examples derived from procedurally generated 3D scenes. Scenes consist of up to four objects, chosen from 59k unique objects, placed on a flat textured surface with an environment lighting map, a three-point lighting system that moves with the camera, and a directional light. As shown in Figure 1, we support four types of object edits: (a) translation across the surface; (b) rotating around the axis orthogonal to the surface; (c) inserting new objects; and (d) removing existing ones. Our 3D rendering engine ensures that all edits are physically plausible and the generated images capture realistic changes in 3D geometry, illumination, and shading resulting from the underlying edit. For instance, rotation and translation require maintaining contact with the surface; inserting new objects requires identifying stable supported poses for new objects; and removing objects often requires rendering occluded objects. Each image contains a language instruction describing one of the four edits and a resulting ground truth edited image. Edited images are evaluated using quantitative metrics that capture realism and faithfulness to the ground truth.

We also introduce 3DIT (**3**D-aware **D**iffusion **I**mage-editing with **T**ext), a model which supports each of the four manipulation tasks with language conditioning. 3DIT is initialized with the Zero-1-to-3 [47] diffusion model (which was trained to perform novel view synthesis) and finetuned on the OBJECT dataset for object-centric image editing. The resultant model has effectively been obtained using a three-stage learning curriculum, starting with massive stable diffusion's web-scale pre-training on image-text pairs, followed by Zero-1-to-3's pre-training stage to enhance the model's understanding of 3D objects, and finally with fine-tuning on OBJECT to enable object-centric edits.

On OBJECT's test images, 3DIT outperforms baselines across all four tasks on metrics that capture the faithfulness of the scene edit. Given the known limitations of automatic quantitative metrics, we also provide a human evaluation study, where 3DIT's outputs are preferred to the baselines over 70% of the time. Edits produced by 3DIT tend to preserve the original scene's structure and not just the edited object. 3DIT preserves the scale and viewpoint of objects, it removes and adds appropriate shadows wherever necessary, and even infills previously occluded portions of the image when the occluder is translated or removed. A multi-task variant of 3DIT performs well despite having to support all four transformations using a single set of parameters. Finally, 3DIT generalizes surprisingly well to new image domains such as CLEVR, a popular synthetic dataset for visual

reasoning, as well as real-world images (see Figure 1). This highlights 3DIT's remarkable capability given that OBJECT is a synthetic, procedurally generated dataset.

## 2 Related work

**Image editing with generative models:** The goals of editing objects and semantic regions in images with language have been active for over a decade [45]. Back then, productizable edits were limited to simple changes like cropping, colorization and resizing to complex procedures such as object removal, addition, and rearrangement [59, 30, 18, 46, 16, 85, 22, 4, 71, 53]. Traditionally, these tasks were performed manually using tools like Adobe Photoshop. However, the origin of Generative Adversarial Networks (GANs) [23] revolutionized the field, propelling significant strides toward automation. StyleGAN [36, 37, 35] notably facilitated intricate modifications to the synthesized images, paving the way for sophisticated GAN-based editing techniques with greater control and flexibility [66, 84, 11, 60, 10, 5, 1, 67]. Since then, advancements in generative image architectures have been marked by the emergence of diffusion models [17]. When coupled with the availability of large-scale image-text datasets [64], these models have facilitated the generation of high-fidelity, diverse scenes [51, 58, 62, 63, 34]. Concurrent with these developments, a new wave of image editing methodologies utilizing these large-scale diffusion models have been introduced [83, 50, 38, 29, 43, 19, 72]. Despite these advancements, models lack the 3D awareness necessary for maintaining geometric and lighting consistency. Our dataset, OBJECT, aims to bridge this gap by enhancing existing methods and serving to evaluate future methodologies.

**3D image editing:** A host of recent research, including StyleNeRF [25], ViewGen [3], EG3D [6], SJC [73], DreamFusion [54], Zero-1-to-3 [47], and Make-It-3D [70], has explored lifting 2D images to 3D. By contrast, our model—3DIT— comprehensively considers the entire scene, not just the object of interest, encompassing geometry, lighting, and other salient attributes of the background.

**Scene rearrangement:** Current research in scene rearrangement tasks primarily involve solving rearrangement from robotic manipulation and embodied agents [39, 55, 56, 48] to provide more intuitive and human-like commands for scene manipulation and navigation. Specific attempts have also been made to apply these techniques to room rearrangements [74, 79, 78] using datasets like AI2-THOR [41], Habitat [69], Gibson [80],3D-FRONT [20] or fine-tuning diffusion models with convex decomposition [72]. For instance, LegoNet [78] focuses on room rearrangements without the need to specify the goal state, learning arrangements that satisfy human criteria from professionally arranged datasets provided by 3D-FRONT [20]. Distinct from these works, our research introduces a unique perspective. We focus on object-level rearrangements with a primary emphasis on 3D-aware image editing using language instructions. 3DIT is trained with OBJECT to edit scenes with a high degree of realism and 3D coherence.

**3D asset datasets:** A diverse set of 3D asset dataset such as ShapeNet [7] and the recent Objaverse [14] have played a pivotal role in 3D computer vision. ShapeNet provides a richly-annotated, large-scale dataset of 3D shapes that has found numerous applications in object recognition, scene understanding, and 3D reconstruction. Objaverse has offered a large collection of 3D objects that are semantically segmented and paired with natural language descriptions. Objaverse has been instrumental in the construction of OBJECT and also advancing several other related research areas, including generating textured meshes [8, 21, 27] zero-shot single image 3D generation [47] and enriching simulators [42, 15] for Embodied AI.

**Synthetic datasets for vision models:** Diagnostic datasets such as CLEVR [33] and CLEV-ERER [81] provide a rigorous test bed for the visual reasoning abilities of models. They contain synthetically generated images of 3D scenes with simple primitives and associated questions that require an understanding of the scene's objects, attributes, and relations to answer correctly. Kubric [24] is an image and video dataset generation engine that can model physical interactions between objects. In a similar vein, OBJECT offers procedurally generated scenes of commonly occurring natural objects derived from ObjaVerse [14] with configurable 3D objects and associated language instructions.

**Benchmarks for image editing:** There is currently a scarcity of benchmarks to evaluate generative models [32], especially for 3D scene editing. Existing ones, including light probes [76], repopulating street scenes [77], GeoSim [9] and CADSim [75] are not publicly available. Our presented OBJECT benchmark will be made publicly available.

# 3   OBJECT: A benchmark for Object Editing in Context

Our goal is to design and evaluate image editing models capable of editing objects in scenes. To enable training and evaluation of such models, we develop OBJECT. OBJECT contains scenes with multiple objects placed on a flat textured surface and illuminated with realistic lighting. These edits are described to the model using a combination of language and numerical values (e.g. pixel coordinates and object rotation angle). All edits result in structural changes to the scene which in turn affect illumination changes such as inter-object reflections and shadows. The model does not have access to the underlying 3D scene (including object segmentations, locations, 3D structure, and lighting direction); it must infer these from the input pixels.

## 3.1   Object editing tasks

OBJECT supports four fundamental object editing tasks: Each of the following manipulations targets a single object within a scene that may contain multiple objects. We now describe each task and the capabilities required from an image editing model to succeed at the task. For specifying locations in an image, we use a coordinate system where (0,0) represents the bottom-left corner and (1,1) the top-right corner. Objects are specified in each task using their crowdsourced descriptions.

**Translation**: Given the x-y coordinates of a target location, a specified object is moved from its original location in the scene to the target location while preserving its angular pose and surface contact. Since the camera is fixed relative to the scene, a change in object location requires to model to synthesize newly visible portions of the object. The model is required to change the object's scale in the image due to perspective projection i.e. the objects should appear smaller when moved further away from the camera and vice-versa. The new location may also result in drastically different illumination of the object.

**Rotation**: A specified object is rotated counter-clockwise around the vertical axis passing through the object's center of mass and perpendicular to the ground by a given angle. To succeed, the model must localize the object, extrapolate the object's shape from a single viewpoint, and re-imagine the scene with the rotated object. Rotating objects leads to intricate changes to the shadow projected on the ground plane which are challenging to accurately produce.

**Insertion**: Given a language description, an object matching the description is added to the scene at a designated x-y location. The model must perform object generation at the desired location with stable pose and surface contact. Besides modeling the object shape, the model also needs to understand the interaction of the geometry with scene lighting to generate a realistic shadow for the object.

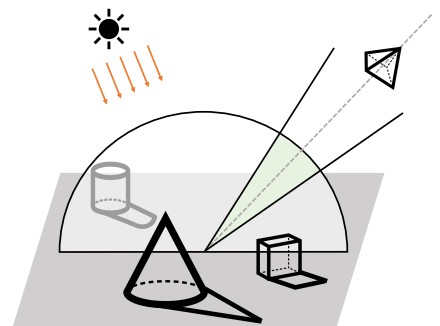

**Removal**: A specified object is removed from the scene. The model must not only be able to locate and segment the object, but also in-paint the object region using scene context. This often requires inpainting an object that was previously partially or fully occluded.

Figure 2: Scene generation in OBJECT depicting camera constraints, directional lighting (environment and three-point lighting not shown), and the resulting object shadows.

## 3.2   Benchmark curation

Paired image-&-text data is plentiful on the internet and large corpora are commonly used to train text-to-image models. However, there is a lack of image editing data consisting of initial and edited image pairs, with a description of the edit. Gathering such a dataset at scale from the real world requires significant manipulation and annotation effort. Our key insight is that while object manipulation data is difficult to acquire, it is much easier to synthesize large volumes of this data leveraging the latest advances in photorealistic rendering and large 3D asset libraries. Therefore, OBJECT contains procedurally generated 3D scenes rendered with objects from these asset libraries.

**Object source**. OBJECT scenes are constructed using one to four 3D objects from the Objaverse dataset[14]. The entire Objaverse dataset contains more than 800k assets. Since objaverse contains objects with errors (some objects are not fully rendered or contain no texture), we filter the objects down to a set of 59k via a combination of Sketchfab metadata-based filtering and crowdsourcing. The resulting objects all have various textures, are easily recognizable, are of high quality and resolution, are free of copyrighted material, are in isolation (as opposed to a single asset with multiple objects), and are free floating (so that they may be placed on any surface in the generated scenes).

Each of these assets is annotated with one of 1613 unique semantic categories using crowdsourcing. Workers were shown a rotating 3D rendering of a particular object and asked to apply a category label; they were provided with a handy autocomplete list of roughly 1400 categories sourced from LVIS [26]categories. If, however, workers were unable to find an appropriate category, they had the option of generating a new category. After this they were asked to write a sentence that describes the object, pointing out any interesting or noteworthy details that would distinguish it from other objects in the same category. Finally, category names were cleaned up to remove spelling errors; we removed unusual or rare categories.

We randomly choose 1513 categories to be seen during training while holding out the remaining 100 as unseen categories for validation and testing. This category split helps quantify the generalization gap in editing previously seen vs novel objects. We use a library of 17 texture maps obtained from [2] to simulate wooden, cobblestone, and brick flooring for the scenes.

**Scene construction**. We limit all scenes to a minimum of one and a maximum of four objects. To identify a natural resting pose for these objects, we perform a physical simulation in Blender where we drop each object onto an XY ground plane and record its resting pose. Then to identify object placements, we sample a bounding box of the same x-y aspect ratio as the object and uniformly scale the object to lie in this bounding box. We ensure that objects, when rotated, do not intersect each other: bounding boxes who's circumscribed circles intersect are rejected. To avoid tiny objects being placed in the same scene as very large objects, we enforce the ratio between the smallest and longest largest side of each bounding box to be greater than $0.8$. We randomly place the camera in the upper hemisphere surrounding the plane and point it towards the origin which lies on the ground plane. We further constrain the camera elevation angle from the ground between $40°$ to $80°$ to ensure that the viewing angle is neither too close to the ground nor completely vertical which are both relatively unnatural. In each scene, there is a designated object that is manipulated. If this object is not visible from the camera, we move the camera away from the origin until the object is visible both before and after the manipulation.

**Scene lighting**. We use several light sources to realistically illuminate the scene. First, we add a random environment lighting map, which are special images that capture the light in a real-world scene from all directions, giving the impression that our constructed scenes are imbedded in various indoor and outdoor locations in the real world. We download 18 of these environment maps with CC0 licences from `https://polyhaven.com/`. Next, we add a three-point lighting system that automatically adapts to the camera view. This involves placing the key light for primary illumination, the fill light to soften key light shadows, and the back light to distinguish the subject from the background. These lights serve to effectively shade the objects in the front of the camera so that their 3D form is apparent. Finally, the scene is illuminated with directional lighting with the direction randomly sampled within a conical neighborhood around the negative-z direction to simulate an overhead light source. This consists of parallel rays emitted by a single light source infinitely far away and therefore can be specified by intensity and direction without specifying a source position.

We generate 100k training examples for each task, and 1024 scenes for validation and testing. The 3D scenes are automatically generated using Blender and its Cycles ray tracer for rendering each scene. We also render segmentation masks that denote object instances, plane and background pixels for all scenes.

## 4   3DIT: a scene-aware editing model

**Task setup.** Consider a 3D scene, $S$, filled with multiple objects. Let $x_1 \in \mathbb{R}^{H \times W \times 3}$ represent an image of this scene produced by a rendering function $f$. Let $l$ represent the text description of the edit, and $v$ represent the task-specific numerial values (i.e. angle for the rotation task and x,y coordinates for removal, insertion, and translation) to describe the desired edit to the scene $S$. In

this paper, we consider object-centric manipulations including rotating, translating, inserting, and removing objects. Manipulating the objects in $S$ can yield a new image $x_2 = f(M(S, l, v))$, where $M$ applied the transformation $l, v$ in 3D.

Our goal is to produce $x_2$ without access to the 3D scene $S$ and instead, directly editing the source image $x_1$. Importantly, we have no explicit information about the scene (including scene geometry and layout), no explicit information about the lighting (such as its location and intensity), and no access to the camera parameters. All this information must be implicitly inferred from the single source image $x_1$. Concretely, we wish to produce the target image $x_2 = \hat{f}_\theta(x_1, l, v)$, where $\hat{f}$ is a learned function with parameters $\theta$.

**Background.** Diffusion models [61] have recently shown spectacular results in generating images conditioned on text descriptions. These models consist of an encoder $\mathcal{E}$ that maps an image $x$ into a latent code $z = \mathcal{E}(x)$, a decoder, $\mathcal{D}$ that can map a latent code back to image space, and a U-Net $\epsilon_\theta$ with learned parameters $\theta$ used for denoising. Some diffusion models are trained on large training corpora such as LAION-5B [65] and are able to produce high-quality high-resolution images that faithfully represent input text descriptions. The recently proposed Zero-1-to-3 model[47] finetunes image-conditioned Stable Diffusion[44] on the task of generating an image of a single object from a novel viewpoint, conditioned on an input view and a relative camera transformation.

**3DIT.** Our model, 3DIT, builds upon Zero-1-to-3. We design $\hat{f}_\theta(\cdot)$ using the same base architecture but make changes to its conditioning module $c_\theta(\cdot)$. Our changes enable the conditioning module to accept edit instructions in the form of language and location information to precisely define the desired edit. In the cross-attention conditional module, Zero-1-to-3 uses a CLIP image encoding to represent the initial image, followed by concatenating a four-dimensional vector encoding camera pose information. This 772-dimensional vector gets passed through a multi-layered perceptron (MLP) to map it back down to a size of 768 dimensions. Similarly, we encode the source image $x_1$ using the same CLIP image encoder. We encode $v$ and concatenated the vector with the image representation and feed it into the MLP. Next, we append the MLP outputs with edit text tokens $l$, which are extracted using CLIP's text encoder.

We finetune our model from the $16,500$-step checkpoint of Zero-1-to-3. During training, the network takes a noised latent encoding of $z_t$, timestep $t$ and conditioning information $c(x_1, l, v)$, where $z_t$ is the latent representation of the target image at time step $t$. and produces a denoising score estimate $\epsilon_\theta(z_t, t, c(x_1, l, v))$ where $c(\cdot) \in \mathbb{R}^{768 \times N}$ outputs a sequence of conditional embedding vectors. We finetune the network with the standard diffusion loss [31, 62]:

$$\min_\theta \mathbb{E}_{z \sim \mathcal{E}_\theta(x_1), t, \epsilon \sim \mathcal{N}(0,1)} ||\epsilon - \epsilon_\theta(z_t, t, c(x_1, l, v))||.$$

# 5 Experiments

We now present experiments to evaluate our 3DIT model. First, we evaluate single task variants of 3DIT, i.e. one model for each of the four tasks – object rotation, translation, insertion and removal. For each of these tasks, we evaluate the performance of the model on novel scenes with objects seen at training time, and with objects unseen at training time. We also provide evaluations for a multi-task model – trained to perform all four tasks.

## 5.1 Baselines

For each of the four tasks, we create strong baselines inspired by recent approaches like VisProg [28] and Socratic models [82] that chain multiple foundation models together to create performant systems for various tasks including image editing.

**Removal:** We first use SAM [40] in the generation mode to get candidate masks for the entire scene and select the mask that contains the point and occupies no more than a third of the area of the entire image. If no such mask is found, we attempt to get a mask by directly using the point as input to SAM to get a mask. Then, we use Stable Diffusion (SD) to inpaint the masked region using the prompt "a rendering of an uncluttered textured floor with no objects".

**Insertion:** This baseline uses SD and the target location to re-imagine the scene with an object of the provided category. The final image is generated by using the prompt "a 3D rendering of category on

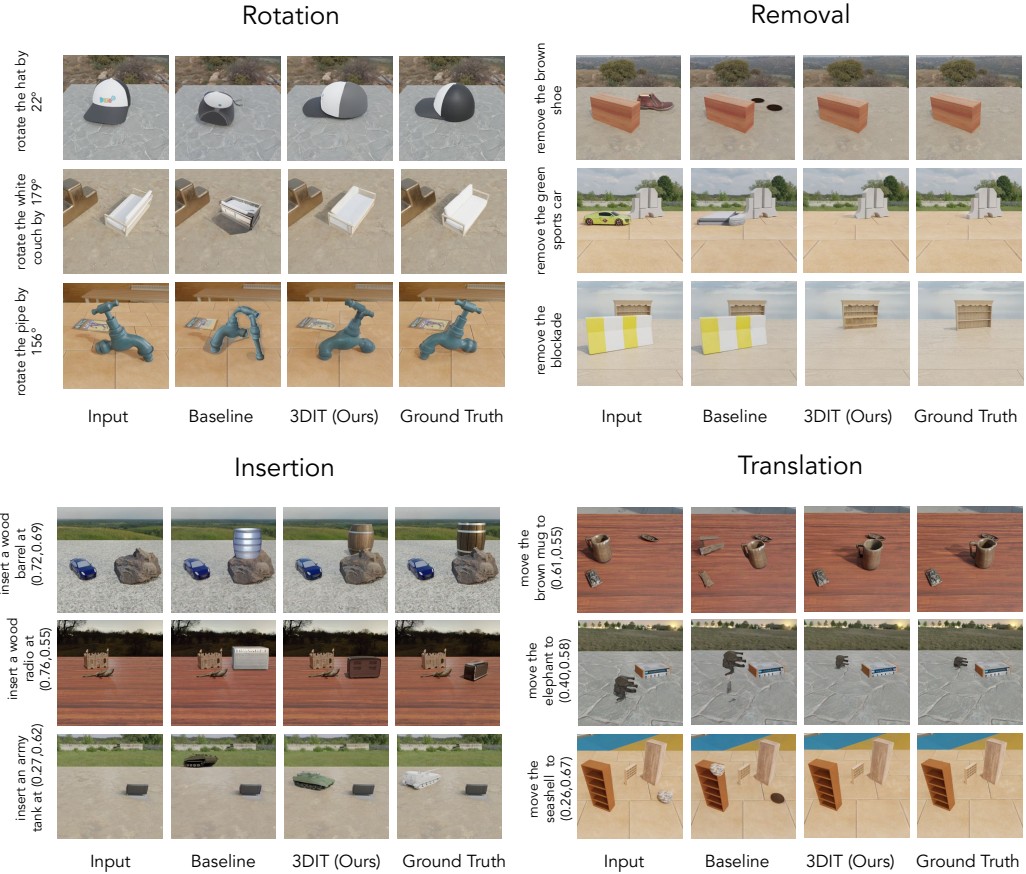

Figure 3: Generated examples from 3DIT as well as baselines for each of the four tasks in the OBJECT benchmark.

a textured floor" conditioned on the initial image and a fixed-size ($200 \times 200$) square mask around the target location.

**Translation:** Translation requires localizing the object given the category name, removing it from the initial location, and inserting it at the target location. We use OWL-ViT [49] to localize the object given the category name. The detected bounding box is fed into SAM to generate the object segmentation mask which is then used for inpainting similar to the Removal baseline. Finally, the segmented object is composited at the target location.

**Rotation:** Here we use Zero-1-to-3 [47] as a baseline which requires the object to be tightly centered in the image with a white background. So, we first localize the object using OWL-ViT, crop the localized region, and segment it using SAM to create the appropriate input for Zero-1-to-3 for performing the rotation. The rotated object is composited back onto the image and the remaining unfilled regions are inpainted using SD.

## 5.2    Quantitative evaluation

We follow Zero-1-to-3 and use four metrics to automatically evaluate the quality and accuracy of the edited image - PSNR, SSIM, LPIPS, and FID. The first 3 directly compare the prediction to the ground truth image, while FID measures the similarity between the predicted and ground truth sets of images. Instead of computing the metrics for the whole image, we focus on the region where the edits are targeted. To do this, we simply use the ground truth segmentation mask to crop the targeted rectangular region of interest prior to computing the metrics. Since our model, as well as our baselines, can generate multiple solutions for each input, our evaluation considers the best-of-four prediction as per the SSIM metric to compute the final scores for all metrics. This considers the

Table 1: Quantitative evaluation using generated samples. For each method, four samples per test image were generated. The best image according to the PSNR metric is selected to represent each sample, and these values are averaged across samples. To ensure that the metrics focus on the transformed object and not the background which mostly remains unchanged, metrics are computed using the region around the transformed object's mask.

| Model | Seen Objects | | | | Unseen Objects | | | |
|---|---|---|---|---|---|---|---|---|
| | PSNR ↑ | SSIM ↑ | LPIP ↓ | FID ↓ | PSNR ↑ | SSIM ↑ | LPIP ↓ | FID ↓ |
| *Task: Translation* | | | | | | | | |
| Baseline | 13.699 | **0.309** | 0.485 | 0.942 | 14.126 | **0.326** | **0.467** | 0.968 |
| 3DIT(1-task) | 14.546 | 0.273 | 0.494 | 0.254 | 14.400 | 0.262 | 0.498 | 0.261 |
| 3DIT(Multitask) | **15.21** | 0.300 | **0.472** | **0.244** | **15.200** | 0.292 | 0.477 | **0.253** |
| *Task: Rotation* | | | | | | | | |
| Baseline | 13.179 | 0.269 | 0.540 | 0.997 | 12.848 | 0.270 | 0.538 | 1.693 |
| 3DIT(1-task) | 16.828 | **0.386** | **0.428** | 0.291 | **16.293** | **0.372** | **0.445** | 0.280 |
| 3DIT(Multitask) | **16.859** | 0.382 | 0.429 | **0.248** | 16.279 | 0.366 | 0.447 | **0.236** |
| *Task: Insertion* | | | | | | | | |
| Baseline | 12.297 | **0.269** | 0.594 | 0.969 | 12.542 | **0.275** | 0.584 | 1.325 |
| 3DIT(1-task) | 13.469 | 0.267 | **0.549** | 0.254 | 12.974 | 0.261 | **0.566** | 0.233 |
| 3DIT(Multitask) | **13.630** | 0.263 | 0.551 | **0.222** | **13.088** | 0.261 | 0.568 | **0.214** |
| *Task: Removal* | | | | | | | | |
| Baseline | 12.494 | 0.383 | 0.465 | 0.801 | 12.123 | 0.379 | 0.459 | 1.047 |
| 3DIT(1-task) | 24.937 | **0.588** | 0.254 | 0.241 | 24.474 | 0.561 | **0.260** | 0.258 |
| 3DIT(Multitask) | **24.980** | 0.585 | **0.249** | **0.236** | **24.661** | **0.568** | **0.260** | **0.240** |

typical use case for editing applications where a user has the flexibility to pick from a range of generated solutions. We report metrics separately for seen and unseen object categories.

Table 1 presents quantitative evaluations for 3DIT in comparison to the baselines. 3DIToutperforms the baselines for all four tasks at the metrics PSNR, SSIM and LPIP. Notably, the multi task model does well in comparison to the single task variant, in spite of having to learn 4 tasks using the same number of learnable parameters. The FID scores for the baseline models tend to be higher. This is because the baselines tend to cut/paste objects in the image (for e.g. in the translation task), which retains image fidelity, even if the scale of the object is incorrect. 3DIT on the other hand does not explicitly cut/paste segments and instead must render them using the diffusion process, and is thus prone to a poorer fidelity. On the contrary, our model is able to properly account for a variety of challenging changes to the underlying 3D scene when editing images, as shown in Figure 4. Its worth noting that the automatic evaluation metrics have limitations and often do not capture editing nuances encompassing geometry, lighting, and fidelity to the instruction. This motivates the need for human evaluation studies.

## 5.3 Human evaluation studies

We conduct human preference evaluations between 3DIT and the relevant baseline by showing two images and asking annotators to select the one that best matches the ground truth image. We measure (1) **Geometric consistency** – This requires humans to consider the geometric correctness of the transformed object, including the scale, positioning of the object on the ground plane and its relationship to other objects. It also requires humans to consider the correctness of other objects in the scene which may get occluded or unoccluded as a result of the transformation. the source caption. (2) **Lighting consistency** – This requires humans to consider the lighting correctness of the transformed object, including the direction and scale of the shadow as a result of the directional lighting. It also requires humans to consider the correctness of the shadows of other objects in the scene which may get occluded or unoccluded as a result of the transformation. Both evaluations also allow a third option (Tie) to be selected. Each pairwise evaluation is carried out for 30 test samples for OBJECT evaluation and 20 samples for real world evaluation.

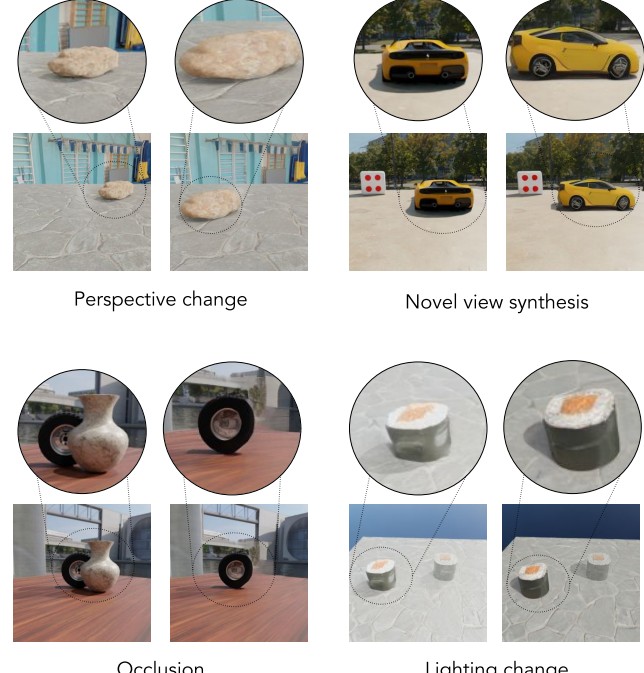

Perspective change      Novel view synthesis

Occlusion      Lighting change

Figure 4: The figure shows the ability of 3DIT to handle various challenges of 3D-aware image editing such as: (a) (*Top left*) perspective size changes; (b) (*Top right*) synthesizing novel view points; (c) (*Bottom left*) generating occluded regions; (d) (*Bottom right*) accounting for scene lighting while rendering objects and their shadows.

Table 2 presents a human evaluation study of the 3DIT model (in a single task setting) in comparison to the corresponding baseline for all four tasks for images from OBJECT and the real world. 3DIT is heavily favored by humans, consistently obtaining preference scores of 70 % and more across all four tasks for geometric as well lighting consistency. The tied scores refer to instances where both models did exceedingly poorly and where both models did a close to perfect job.

For the translation task, 3DIT is able to scale the object appropriately, as well rendering the shadow correctly. The baseline, in particular, does a poor job of the shadow and gets the scale wrong, leading to a physically implausible image. For the rotation task, 3DIT performs a rotation consistent with the ground plane and also renders a superior shadow. For the removal task, 3DIT tends to inpaint occluded objects well, and correctly adjusts their shadows. It also does well at removing the entire extent of the correct object in contrast to the baseline.

### 5.4 Real-world transfer qualitative

While we train our models on simulated data, we test the model's ability to transfer to real-world images qualitatively. Figure 5 shows our model's output for different prompts for the same input image for all four tasks. We find these preliminary results encouraging as the outputs not only respect the task description but also look reasonably photo-realistic with appropriate shadows despite never seeing real-world editing examples during training.

## 6 Limitations and Broader Impact

Our work explores the use of synthetic data for training physically plausible and scene-aware image editing models. Given that even training on scenes with limited realism and complexity results in models that transfer well to the real world, there is tremendous potential to significantly improve performance by using more advanced photo-realistic simulators. We give an analysis of the typical failure cases of our model in the appendix. Finetuning on a small set of hand-crafted real-world editing examples may also improve transfer to real-world images and enable compelling editing applications. Our work leads the way towards easy-to-use and increasingly powerful image editing

Table 2: Outcome of the human evaluation. The table illustrates the evaluators' preferences for 3DIT assessed on geometric accuracy and 3D lighting consistency. Baseline methods rarely gained preference due to their limited capacity to maintain geometric quality and lighting consistency.

| | OBJECT evaluation | | | | | |
|---|---|---|---|---|---|---|
| Task | Geometric consistency | | | Lighting consistency | | |
| | Baseline | 3DIT*(Ours)* | Tie | Baseline | 3DIT*(Ours)* | Tie |
| Translation | 20.0 % | 73.3 % | 6.6 % | 3.3 % | 80.0 % | 16.6 % |
| Rotation | 3.3 % | 80.0 % | 16.6 % | 6.6 % | 73.3 % | 20.0 % |
| Insertion | 13.3 % | 70.0 % | 16.6 % | 10.0 % | 73.3 % | 16.6 % |
| Removal | 3.3 % | 86.6 % | 10.0 % | 0.0 % | 86.6 % | 13.3 % |
| | Real world evaluation | | | | | |
| Task | Geometric consistency | | | Lighting consistency | | |
| | Baseline | 3DIT*(Ours)* | Tie | Baseline | 3DIT*(Ours)* | Tie |
| Translation | 25.0 % | 60.0 % | 15.0 % | 15.0 % | 70.0 % | 15.0 % |
| Rotation | 10.0 % | 80.0 % | 10.0 % | 10.0 % | 80.0 % | 10.0 % |
| Insertion | 35.0 % | 55.0 % | 10.0 % | 35.0 % | 55.0 % | 10.0 % |
| Removal | 10.0 % | 75.0 % | 15.0 % | 5.0 % | 80.0 % | 15.0 % |

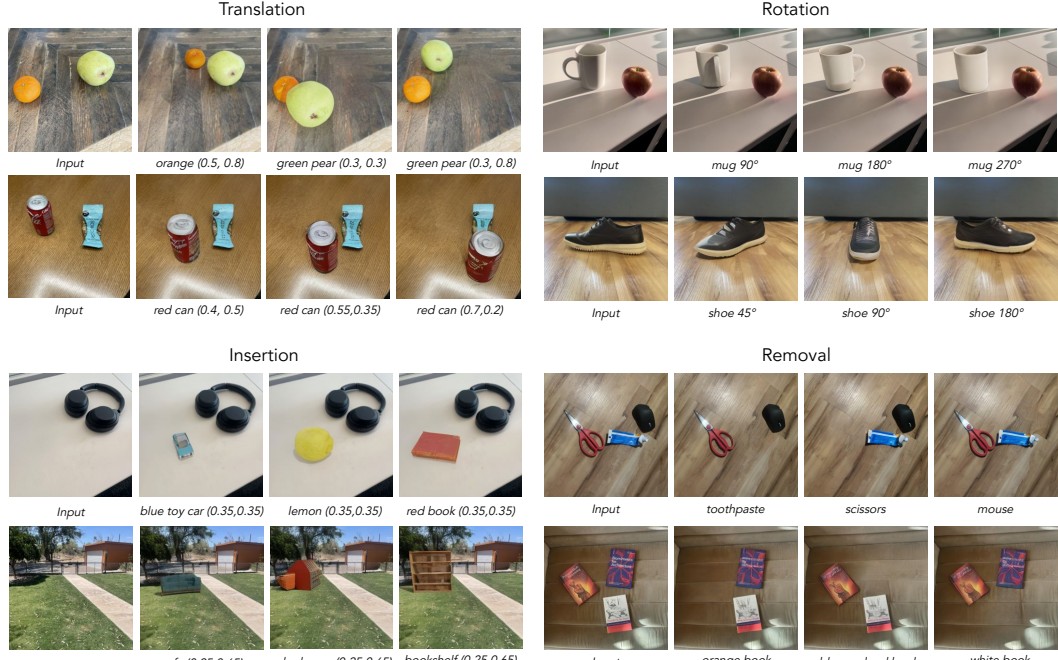

Figure 5: 3DIT is able to generalize to the real world while only being trained on a synthetic dataset. Here we show varying prompts for each of the four editing tasks.

capabilities for the broader society in the near future. Like any generative model, our work could also potentially be misused for propagating misinformation.

# 7   Conclusion

This work presents 3DIT, a model capable of editing individual objects within images, given a language instruction. 3DIT is trained on a new dataset, OBJECT, consisting of 400k 3D scenes procedurally generated using Objaverse objects. 3DIT performs well across on OBJECT and shows promising generalization to CLEVR as well as the real world.

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

# 8 Appendix

## 8.1 Training and Inference Details

We closely follow the training procedure established by [47], with a few modifications. Our approach uses an effective batch size of $1024$, which is smaller than the batch size of $1536$ used by Zero-1-to-3. This adjustment was necessary because of the additional memory requirements caused by the reintroduction of the CLIP text encoder. This batch size is achieved by using a local batch size of $64$ across 40GB NVIDIA RTX A6000 GPUs, along with two gradient accumulation steps. Similar to Zero123, we train on images with a resolution of $256 \times 256$, resulting in a latent spatial dimension of $32 \times 32$. Following their protocol, we utilize the AdamW optimizer, with a learning rate of 1e-4 for all parameters of the model except for those of the concatenation MLP, which uses a learning rate of $1e-3$. Our training process runs for a total of 20,000 steps. We then select the best checkpoint based on our metrics computed from an unseen object validation set. As was the case in StableDiffusion, we freeze the CLIP text encoder during training. For inference, we generate images with the DDIM [68] sampler using 200 steps. We do not use classifier-free guidance, i.e. the cfg term is set to 1.0.

## 8.2 Robustness to severity of transformation

We analyze the robustness of our method by measuring the performance of the single task rotation model as the complexity of the scene and severity of transformation changes. In Figure 6, we show the average of our Mask PSNR metric as the number of objects in the scene varies from $1$ to $4$, where a slight drop in performance occurs as the number of objects increases. In Figure 7, we show average Mask PSNR for rotations in a given angle range on a pie chart, where it can be seen that the model does better with smaller angle deviations.

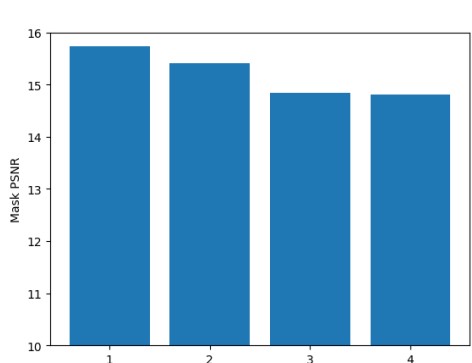

Figure 6: Average Mask PSNR of the single-task rotation model as the number of objects in the scene varies.

Figure 7: Average Mask PSNR of the single-task rotation model for rotation angles falling within a slice.

## 8.3 Model failure analysis

Here we analyze the most common failure model of 3DIT.

**Incorrect geometry:** The model incorrectly changes the object's geometry.

**Unintended global modifications:** The model changes parts of the scene that are not intended to be edited.

**Incorrect texture:** The model changes details on the surface of the object, failing to preserve its identity.

**Incorrect localization:** The model performs the requested task, but on an object that is different from the intended one.

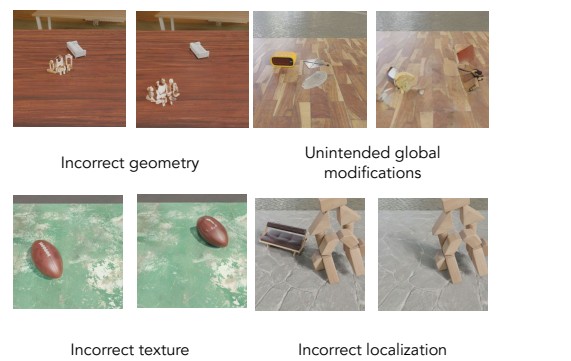

Incorrect geometry          Unintended global
                            modifications

Incorrect texture           Incorrect localization
                            (instruction: remove block)

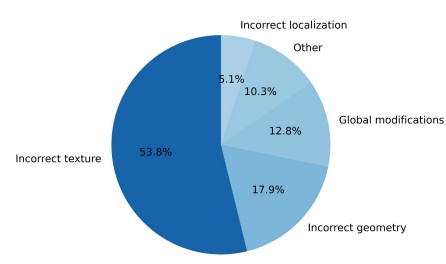

Figure 8: Qualitative examples of the four must common failure cases produced by the model.

Figure 9: Frequency of most common failure cases. We examine 50 random outputs of the model and bucket the model errors by failure type

## 8.4 Initialization ablation

To test the effect of the model's initialization, we run an ablation training our model on three different initializations: Image-conditioned Stable Diffusion [44], Zero123 [47] (our standard initialization), and the recently released Zero123-XL [13], which is a Zero123 model trained on a much larger version of the Objaverse dataset with 10M assets. We see that although Zero123-XL was is a more performant base model for the task of rotating a single object, it does not cause an improvement overall on the tasks we consider here.

Table 3: Quantitative results for training our single-task model across three different ablations: Image-conditoined Stable Diffusion, Zero123, and the recently released Zero123-XL.

| Model | Seen Objects | | | | Unseen Objects | | | |
| | PSNR ↑ | SSIM ↑ | LPIP ↓ | FID ↓ | PSNR ↑ | SSIM ↑ | LPIP ↓ | FID ↓ |
|---|---|---|---|---|---|---|---|---|
| | *Task: Translation* | | | | | | | |
| SD | 14.373 | 0.264 | 0.510 | **0.101** | 14.351 | 0.253 | 0.510 | **0.102** |
| Zero123 | **15.210** | **0.300** | **0.472** | 0.244 | **15.200** | **0.292** | **0.477** | 0.253 |
| Zero123-XL | 15.121 | 0.294 | 0.477 | 0.252 | 15.052 | 0.286 | 0.478 | 0.239 |
| | *Task: Rotation* | | | | | | | |
| SD | 15.074 | 0.368 | 0.430 | **0.089** | 14.558 | 0.359 | 0.438 | **0.095** |
| Zero123 | **16.859** | **0.382** | **0.429** | 0.248 | **16.279** | 0.366 | 0.447 | 0.236 |
| Zero123-XL | 15.433 | 0.381 | 0.420 | 0.241 | 15.008 | **0.378** | **0.429** | 0.243 |
| | *Task: Insertion* | | | | | | | |
| SD | 13.220 | 0.253 | 0.570 | **0.108** | 13.131 | 0.255 | 0.572 | **0.100** |
| Zero123 | **13.630** | 0.263 | **0.551** | 0.222 | 13.088 | **0.261** | 0.568 | 0.214 |
| Zero123-XL | 13.481 | **0.264** | 0.557 | 0.274 | **13.094** | 0.259 | **0.566** | 0.258 |
| | *Task: Removal* | | | | | | | |
| SD | 23.882 | 0.576 | 0.263 | **0.117** | 23.352 | 0.542 | 0.270 | **0.115** |
| Zero123 | **24.980** | **0.585** | **0.249** | 0.236 | 24.661 | 0.568 | 0.260 | 0.240 |
| Zero123-XL | 24.775 | 0.585 | 0.255 | 0.247 | **24.830** | 0.568 | **0.253** | 0.215 |

## 8.5 Additional qualitative results

In Figure 10, we show qualitative results from our multitask model on each of the four editing tasks.

Table 4: Summary of key statistics of the OBJECT dataset.

| | |
|---|---|
| Total objects | 62950 |
| Total categories | 1613 |
| Object per category median | 6.0 |
| Object per category mean | 39.03 |
| Object per category std | 138.00 |

## 8.6 Dataset Analysis

In this seciton, we provide some details about the composition and statistical makeup of our dataset. In Table 4, we show a statistical overview of the dataset, including total number of objects and categories, as well as the mean, median, and standard deviation of objects per category. We also visualize the distribution of objects across categories, as can be show in the histogram in Figure 11. Finally, we visualize the frequency of category names in the wordcloud in Figure 12.

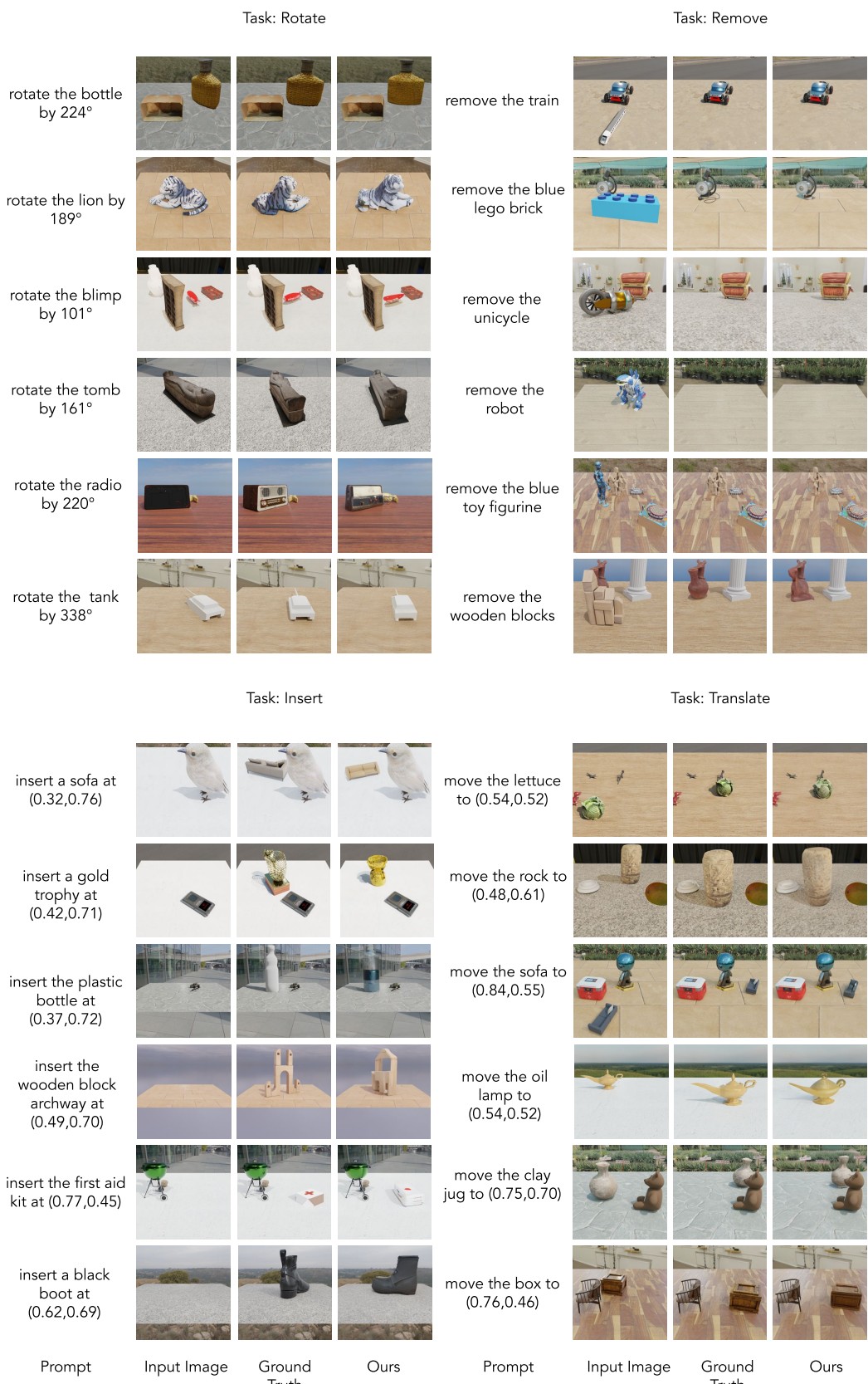

Figure 10: Generated examples by the 3DIT multitask model.

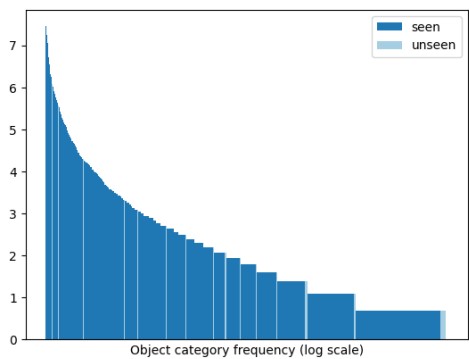

Figure 11: Object categories from seen and unseen splits sorted by frequency, in log scale.

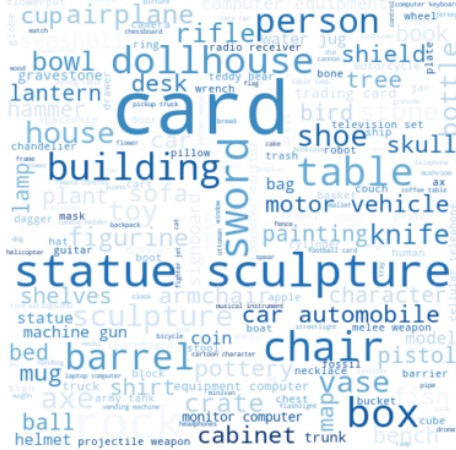

Figure 12: A wordcloud visualizing the frequency of various object category names.

