# OpenReview forum: "OBJECT 3DIT: Language-guided 3D-aware Image Editing"
_NeurIPS.cc/2023/Conference — NeurIPS 2023 poster_

### Official Review · Reviewer_SoaB · 2023-06-30

**Soundness:** 3 good
**Presentation:** 3 good
**Contribution:** 3 good
**Rating:** 6
**Confidence:** 4

**Summary:**

The paper studies the problem of object-centric image editing. The authors first curate a dataset based on Objaverse by selecting high-quality textured samples, and then simulate+render them on a plane. The objects can be manipulated in 3D and rendered correspondingly, which generates the groundtruth for training learning-based object editing models. The paper further presents a diffusion-based object editing model (3DIT) based on zero-123, where the major difference is the addition of editing prompt conditioning. Results in quantitative and qualitative experiments show 3DIT outperforms baselines based on foundation models.

**Strengths:**

The dataset curation using 3D simulation for the object-centric image editing task makes sense. This task intrinsically requires understanding of the 3D world and 3D-aware image formation process. The proposed dataset is guaranteed to be 3D-correct, and would be useful for research along this direction. The proposed method also achieved great performance compared to the baselines. Besides, the paper is well-written and easy to follow.

**Weaknesses:**

1. The realism of the generated dataset is still limited:
a) The single directional light (why not multiple lights or Image-based lighting?) which makes the shadow and shading's distribution not diverse/realistic enough.
b) The size range of the object is quite limited (0.8 ratio threshold), in practice, there are a lot of cases where different sized objects are placed nearby (also the true size seems not kept, so chairs and lamps appears to be of similar scale).
The authors indeed show some qualitative results of direct transfer to real data, but no 1) quantitative evaluation of sim2real or 2) comparison to baselines' generalization have been provided.

2. Although the data is generated with 3D, the method follows a 2D design. Such a model does not exhibit a great understanding of 3D objectness -- for example, the identity of the object is sometimes not well-kept during rotation or translation (e.g. the change of coke/headphone texture in Fig 1), or the cast shadow or the shading do not make full sense w.r.t. the whole scene if we look close enough (e.g. first 3 rows of clevr in Fig 1). Having some systematic analysis of the failure cases here would have been very helpful for future research.

Minor questions/comments:
1. Why is the background always black? This seems to be creating domain gap, as usually we don't have pure black background in real life.
2. Wrong number highlight in Table 1, 3DIT(Multitask) - insertion - LPIP, 0.585 is worse than the other two.
3. The multi-task model is much worse in terms of FID (Table 1), but not other metrics. Are there any specific reason for this?


**Questions:**

Please see my questions in the weaknesses. Overall, I think the pros of this paper outweigh the cons -- the curated dataset would be useful for future exploration along this direction and the proposed method can serve as a strong 2D baseline. I believe having sim2real comparison and systematic failure case analysis will further solidify the paper.

**Limitations:**

More analysis on failure cases and limitations will be beneficial.

---

> ### Author Rebuttal · Authors · 2023-08-10
>
> We thank the reviewer for taking the time to read our submission and for providing valuable feedback. We will now respond to the highlighted questions and concerns.
>
> **Realism of dataset**
>
> As per reviewer suggestions, we have improved the realism of our dataset in 3 ways: (i) In line with common practices in professional film and photography, we implemented a 3-point lighting system that automatically aligns itself with the viewpoint of the camera, thereby shading objects in a way that better reveals their true 3D form; (ii) We added real-world environmental lighting into our dataset with 360-degree HDRI captures from both indoor and outdoor scenes under a variety of lighting conditions. Not only does this give scenes in our dataset realistic backgrounds, but the light emanated from these captures is integrated into the ray-tracing process during rendering, so that all aspects of the scene benefit from a more realistic lighting distribution; and (iii) We replaced the ground textures in our dataset with more realistic ones that have normal, roughness and displacement maps. Please see figure B in the dataset for a visualization of the improvements.
>
> We also retrained our models on this more realistic dataset and find that the in-domain performance follows similar trends as reported in the paper, but the models generalize better to out-of-domain data like CLEVR and the real world images. We will update the draft with all recomputed metrics for our new models on this improved dataset but include PSNR here for brevity.
>
> |                  | PSNR (seen objects) | PSNR (unseen objects) |
> |------------------|---------------------|-----------------------|
> | Translation      |                     |                       |
> | Baseline         | 13.699              | 14.126                |
> | 3DIT (1-task)    | 14.546              | 14.4                  |
> | 3DIT (Multitask) | 15.21               | 15.2                  |
> | Rotation         |                     |                       |
> | Baseline         | 13.179              | 12.848                |
> | 3DIT (1-task)    | 16.828              | 16.293                |
> | 3DIT (Multitask) | 16.859              | 16.279                |
> | Insertion        |                     |                       |
> | Baseline         | 12.297              | 12.542                |
> | 3DIT (1-task)    | 13.469              | 12.974                |
> | 3DIT (Multitask) | 13.63               | 13.088                |
> | Removal          |                     |                       |
> | Baseline         | 12.494              | 12.123                |
> | 3DIT (1-task)    | 24.937              | 24.474                |
> | 3DIT (Multitask) | 24.98               | 24.661                |
>
> **2D design**
>
> Our goal is to see if it is possible to perform 3D-aware edits while working solely in the pixel space. This hypothesis is motivated by the success of approaches like Zero-123 which generate plausible novel views of a single object given only one image. Importantly, by using a large pretrained image generator, Zero123 outperforms comparable methods that use explicit 3D geometry. Our results further provide evidence to support that performing 3D-aware edits while solely operating in 2D is possible.
>
> **Failure cases analysis**
>
> We conducted an analysis of the model's failure modes and report 4 major categories: incorrect texture, incorrect localization, unintended global modifications and incorrect geometry. Figure C in the attached PDF shows the frequency of each error type among a sample of errors annotated from our test set. Figure D shows a visualization of each error category.
>
> **Human eval**
>
> We also provide quantitative human-evaluation on real images. We find that human evaluators overwhelmingly prefer results from our method over baselines. Among all tasks, insertion is the most challenging for our model.
>
> | Task        | Geometric consistency |             |     | Lighting consistency |             |     |
> |-------------|-----------------------|-------------|-----|----------------------|-------------|-----|
> |             | Baseline              | 3DIT (Ours) | Tie | Baseline             | 3DIT (Ours) | Tie |
> | Translation | 25%                   | 60%         | 15% | 15%                  | 70%         | 15% |
> | Rotation    | 10%                   | 80%         | 10% | 10%                  | 80%         | 10% |
> | Insertion   | 35%                   | 55%         | 10% | 35%                  | 55%         | 10% |
> | Removal     | 10%                   | 75%         | 15% | 5%                   | 80%         | 15% |
>
> Below we compare the model trained on the new improved dataset and the old dataset. Training on the newer more realistic dataset leads to stronger models.
>
> | Task        | Geometric consistency |           |     | Lighting consistency |           |     |
> |-------------|-----------------------|-----------|-----|----------------------|-----------|-----|
> |             | Old model             | New model | Tie | Old model            | New model | Tie |
> | Translation | 15%                   | 60%       | 25% | 15%                  | 65%       | 20% |
> | Rotation    | 30%                   | 55%       | 15% | 15%                  | 60%       | 25% |
> | Insertion   | 30%                   | 60%       | 10% | 20%                  | 50%       | 30% |
> | Removal     | 10%                   | 60%       | 30% | 10%                  | 60%       | 30% |

---

> > ### Comment · Reviewer_SoaB · 2023-08-13
> >
> > I thank the authors for their detailed response. Most of my concerns are resolved, and I remain on the positive side.

---

> > > ### Author Response · Authors · 2023-08-20
> > >
> > > Thank you very much for taking the time to review our work and for giving valuable feedback. We are glad you found our work to be a positive contribution.

---

### Official Review · Reviewer_eZ2Z · 2023-07-03

**Soundness:** 3 good
**Presentation:** 3 good
**Contribution:** 3 good
**Rating:** 5
**Confidence:** 3

**Summary:**

This paper constructs a dataset containing 400K examples, which is used for the task of language-guided 3D-aware image editing. This paper also proposes a model, named 3DIT, to solve this task. The model is based on 2D diffusion model, which first goes through the pre-training of text-to-image generation and Zero-1-to-3, and then is fine-tuned on the dataset for the 3D-aware image editing task. The model is evaluated on the proposed dataset and achieves state-of-the-art performance.

**Strengths:**

1. This paper constructs a large-scale dataset, which is a good contribution. And language-guided 3D-aware image editing is an important task.
2. The paper is well written and easy to understand.
3. The experiments show that the proposed model trained on the proposed dataset have a reasonable ability on the task of language-guided 3D-aware image editing.

**Weaknesses:**

My biggest concern is that the performance of the proposed model is not very good.

1. The GIF results in the supplementary material exhibit incorrect results on the shadow.
2. The lighting and shadow of experimental results on real-data is not realistic. The quality of edited images degrade a lot.

Due to the poor performance, I am not sure if the quality and diversity of the proposed dataset is good enough.

The illumination and background of presented real data is too simple. It would be great to add more challenging test data to see the upper bound of the proposed model.

**Questions:**

Please see the weakness section.

**Limitations:**

The authors have discussed the limitations.

---

> ### Author Rebuttal · Authors · 2023-08-10
>
> We thank the reviewer for taking the time to read our submission and for providing valuable feedback. We will now respond to the highlighted questions and concerns.
>
> **Incorrect shadow behavior in GIF, dataset realism and diversity**
>
> The original dataset at the time of submission used a single directional light source leading to unrealistic lighting in our training dataset. Therefore, learning to model shadows on OBJECT dataset showed limited transfer to CLEVR. As discussed in the common statement, we have improved the realism of the dataset. Please see Figure B for a visualization of the lighting improvements. Specially, with our new 3-point and environmental lighting, the scenes are now lit in a much more realistic manner leading to noticeably better shadow rendering behavior in our models. See the Figures A and I in the pdf for comparison between models trained on old and new datasets.
>
> **Difficulty of real data**
>
> Figure G in the attached PDF shows the model editing more complex real world scenes, as well as some failure cases. For example, when removing the can, we see some noticeable infilling artifacts. Our model often struggles with scenes of higher complexity in the real world. We also conducted an analysis of the model's failure modes and report 4 major categories: incorrect texture, incorrect localization, unintended global modifications and incorrect geometry. Figure C in the attached PDF shows the frequency of each error type among a sample of errors annotated from our test set. Figure D shows a visualization of each error category.

---

> > ### Comment · Reviewer_eZ2Z · 2023-08-18
> >
> > Thanks for the rebuttal.
> >
> > Although the performance is limited, I would keep the positive side, considering that contribution of the dataset.
> >
> > Please discuss the failure cases in the camera ready version.

---

> > > ### Author Response · Authors · 2023-08-20
> > >
> > > Thank you very much for taking the time to review our work and for giving valuable feedback. We will add a failure analysis to the final paper.

---

### Official Review · Reviewer_yCLA · 2023-07-06

**Soundness:** 3 good
**Presentation:** 3 good
**Contribution:** 2 fair
**Rating:** 5
**Confidence:** 3

**Summary:**

The paper formulates a task of 3D aware editing using the language guidance. The task aims to insert, remove, translate or rotate objects in a scene (2D images) by maintaining the details like shadows, 3D consistency of the object, changes in the object sizes due to perspective projections etc. The model is based on Stable diffusion, Zero 1-to-3 method and fine tuning on the given dataset having editing information and text to describe the edit. The paper promises to release the dataset OBJECT derived from Objaverse, which the authors use to train their model on.  The results in the teaser figure and others show that the model is able to perform the given edits, while preserving the semantics of the image. For examples, the objects are translated and it respects the perspective projection, shadows and placement on the surface. The authors also claim that the method is generalizable to the real images.

**Strengths:**

1) The paper is able to show that the manipulations possible with the method can preserve the 3D properties of the scene including localization of the objects, scaling, shadows and consistency of the inpainted regions.

2) The paper compares with the image based baselines, for examples uses SAM to segment the images and translates the objects in the scene. It also compares with the a 3D baseline for rotation using Zero 1-to-3.

3) The paper shows quantitative and qualitative results of their method comparing scores such as PSNR, SSIM, LIPIPS and FID between the original and edited images. The authors also conduct a user study to access the quality of lighting and geometry.

**Weaknesses:**

1) The method to train on the given dataset is not clear to me. There is no pipeline figure to explain the stages of the training. The first two steps are previous works. The contribution which is in the third step is not explained properly in the paper. Is this fine-tuning stage similar to Zero-1-to-3? How was the editing sequence fed to the network? How is it 3D aware besides the Zero-1-to-3 training? Is the method's 3D consistency (for example in rotation) upper-bounded by Zero-1-to-3?

2) While the images shown in the paper show that some properties are preserved as the editing operations are done, the results in the gifs show some obvious flickering artifacts which do not respect the properties like shadows. This does not go well with the objective of the paper. Besides the problems with Stable Diffusion , where do the problems arise?

3) How is the quality drop if number of sequential operations are done? For example once can perform insertion-> rotation ->translation etc for the same/ different objects in the scene. How does the quality drop compare with the Stable diffusion image editing methods? This is more interesting to me. A 3D aware editing  framework should be able to handle multiple sequential edits with consistent results.

4) Another baseline would to use a monocular depth estimation model (eg Zoedepth, Midas) to extract the surface and perform the edits using Zero-1-to-3. This can handle the perspective projection of the objects, and/or even lighting and shadows. Did the authors try similar more strong baselines? How do they compare with the current method?

5) The paper claims to generalize to the real scenes. This is a significant claim and needs to be evaluated. The issue of synthetic and real image domain gap is an active research area. How does the current method solve that in this particular task? Were real images considered in Table 2?

**Questions:**

Please refer to the weakness section.

**Limitations:**

The authors do not discuss the limitations of their work in detail. Please add a detailed section on where the method fails.

---

> ### Author Rebuttal · Authors · 2023-08-10
>
> We thank the reviewer for taking the time to read our submission and for providing valuable feedback. We will now respond to the highlighted questions and concerns.
>
> **Method clarifications**
>
> Our approach extends zero123 with a CLIP text encoder (the same as the one used in the original StableDiffusion). The model is finetuned starting from the zero123 checkpoint in exactly the same way as zero123. A sample in a minibatch consists of the original image, instruction, and the target image. The denoising network is a  diffusion model trained to recover the target image from a noise version of the target image conditioned on the original image and the instruction. The instruction is always a single edit from one of the four supported tasks. We do not train on sequences. Please see Figures E and F in the attached PDF for further clarification.
>
> We clarify that 3D awareness has to do with the task definition which requires models to edit objects in complex scenes as if performing 3D transformations in the real world but while only operating in the pixel space. Zero-123 is a model that takes an image and generates novel viewpoints of the object. In doing so, Zero-123 has developed an implicit understanding of the 3D geometry of objects. However, Zero-123 does not take into account scene lighting and occlusions with other objects in the scene. Therefore, our model trained on our 3D-aware editing benchmark learns more about the geometry, lighting, and surface-contact than Zero123.
>
> While our current method may be bounded by the limitations of Zero123, the benefit of using this approach is that, unlike approaches that involve explicit 3D geometry, our approach can be improved with scale [1]. Extensions of our framework can develop improved datasets with more objects [1], language annotations [2], or more complex environments [3], for example.
>
> **Flickering artifacts**
>
> The reviewer should also note that our approach is primarily geared towards editing images and not videos and hence the frames generated when creating a video visualization of the results are not guaranteed to be temporally consistent. Ensuring this temporal consistency is an active area of research [4] and would require several innovations in video modeling. Current state of the art video models still struggle with temporal consistency despite being explicitly trained for this [5]. Our newest model has improved in its ability to synthesize shadows for out of domain objects. Please see Figure I in the PDF.
>
> **Sequential operations**
>
> Performing a sequence of edits is an interesting idea! We provide an example of performing multiple edits on the same object in the attached PDF. Please see Figure H.
>
> **Additional baseline**
>
> Zero123 is an image-to-image model conditioned on a camera pose. Zero-123 doesn’t produce a 3D model nor is it conditioned on one. Therefore any 3D surface extracted with a depth estimation model would have to be converted to an image before being processed by Zero123. The baseline reported in our paper uses Zero123 to rotate an object extracted from a segmentation model and then harmonizes the resulting edit with StableDiffusion inpainting. However, in the spirit of your suggestion, to test the maximum possible performance of a method involving Zero123, we ran an additional version of this baseline where the edited object is extracted with its ground truth mask before being rotated by Zero123.  The results are reported below.
>
> |                 | Seen   |        |       |       | Unseen |       |       |       |
> |-----------------|--------|--------|-------|-------|--------|-------|-------|-------|
> |                 | PSNR   | SSIM   | LPIP  | FID   | PSNR   | SSIM  | LPIP  | FID   |
> | Ours multitask  | 16.859 | 0.382  | 0.429 | 0.248 | 16.293 | 0.372 | 0.445 | 0.28  |
> | Oracle Baseline | 15.613 | 0.3845 | 0.422 | 0.248 | 14.552 | 0.366 | 0.439 | 0.001 |
>
> **Real-world evaluation**
>
> To address this concern, we have conducted a human evaluation of real world images. We compare our latest model to the baselines and also compare our latest model to the original one.  Please see the tables below.
>
> | Task        | Geometric consistency |             |     | Lighting consistency |             |     |
> |-------------|-----------------------|-------------|-----|----------------------|-------------|-----|
> |             | Baseline              | 3DIT (Ours) | Tie | Baseline             | 3DIT (Ours) | Tie |
> | Translation | 25%                   | 60%         | 15% | 15%                  | 70%         | 15% |
> | Rotation    | 10%                   | 80%         | 10% | 10%                  | 80%         | 10% |
> | Insertion   | 35%                   | 55%         | 10% | 35%                  | 55%         | 10% |
> | Removal     | 10%                   | 75%         | 15% | 5%                   | 80%         | 15% |
>
> | Task        | Geometric consistency |           |     | Lighting consistency |           |     |
> |-------------|-----------------------|-----------|-----|----------------------|-----------|-----|
> |             | Old model             | New model | Tie | Old model            | New model | Tie |
> | Translation | 15%                   | 60%       | 25% | 15%                  | 65%       | 20% |
> | Rotation    | 30%                   | 55%       | 15% | 15%                  | 60%       | 25% |
> | Insertion   | 30%                   | 60%       | 10% | 20%                  | 50%       | 30% |
> | Removal     | 10%                   | 60%       | 30% | 10%                  | 60%       | 30% |
>
> References
> [1] Objaverse-XL: A Universe of 10M+ 3D Objects
> [2] Scalable 3D Captioning with Pretrained Models
> [3] Habitat: A Platform for Embodied AI Research
> [4] TokenFlow: Consistent Diffusion Features for Consistent Video Editing
> [5] VideoFusion: Decomposed Diffusion Models for High-Quality Video Generation

---

> > ### Comment · Reviewer_yCLA · 2023-08-18
> >
> > While the authors did address some of the concerns, I still believe that there are some quality concerns. That being said I think the dataset is a valuable addition and hence I am raising my rating to borderline accept.

---

> > > ### Author Response · Authors · 2023-08-20
> > >
> > > Thank you very much for taking the time to review our work. Your insightful feedback and experiment suggestions have led us to new findings that have strengthened our work. We greatly appreciate your improved rating.

---

### Official Review · Reviewer_Rpi6 · 2023-07-07

**Soundness:** 3 good
**Presentation:** 4 excellent
**Contribution:** 3 good
**Rating:** 5
**Confidence:** 5

**Summary:**

The authors propose a large dataset of  3D aware image edits along with editing instructions built on the objaverse dataset. They also introduce a model finetuned on Zero-1-to-3 for 3D aware editing tasks which include object insertion, removal, translation and rotation. Comparisons are provided against state of the art models for each task and performance improvement is demonstrated.

**Strengths:**

1. **Clarity**: The paper is well written with attention to detail. All the necessary details particularly with regards to the dataset creation have been adequately explained.
2. **Interesting dataset**: The 400k dataset of images along with edit instructions would serve as an interesting training and benchmarking dataset for the task of 3D aware editing.
3. **Quantitative metrics**: A number of qualitative comparisons and user studies are provided to demonstrate the geometric consistency of the edits and lighting consistency.


**Weaknesses:**

1. **Novelty**: Although the proposed dataset represents an important contribution, the proposed approach relies on zero-1-to-3 and finetuning on a new dataset.
2. **Need for zero-1-to-3**: The approach finetunes a model on top of zero-1-to-3 to incorporate edit instructions. Can the finetuning be done on top of base SD?. Adding an ablation to this effect will be helpful to demonstrate the need for the 3 stage curriculum.
3. **Related work**: Several related work that may provide important context are missing. The authors might find some of the following works relevant and interesting [1,2,3,4]. Although some of these works are pre-prints and do not warrant strict comparisons, incorporating them into the related work section would place the proposed work appropriately w.r.t the landscape of current literature
4. **Changes in the edited image** : There are certain global changes in the edited image, that dilutes some of the claims w.r.t editing. Particularly, for the CLEVR dataset, the provided supplm examples show changes in the color of certain objects upon insertion/ removal


[1] ControlNet
[2] InstructPix2Pix
[3] InstantBooth
[4] GLIGEN

**Questions:**

1. For translation and insertion, what are the location? are they provided in pixel space? Since they are 3D aware, does it make more sense to provide edit instruction in camera coordinates? ( In pixel coordinates, how is depth interpreted?)
2. Is it easier to provide control inputs as a spatial map (say as gaussians/ keypoints on a 2D image) rather than as a pixel locations to allow for more local specificaition?

**Limitations:**

Adequate treatment of limitations have been provided.

---

> ### Author Rebuttal · Authors · 2023-08-10
>
> We thank the reviewer for taking the time to read our submission and for providing valuable feedback. We will now respond to the highlighted questions and concerns.
>
> **Novelty**
>
> There are 3 main novel contributions in our work:
> 1. We propose the task of language-guided 3D-aware image editing.
> 2. Given the challenges involved in creating supervised training data for this task in the real world,  we propose a large-scale procedurally generated benchmark as a means for learning and evaluating this new task.
> 3. We train a model to edit scenes in context and observe real-world transfer.
>
> Identifying a simple approach (adapting Zero123 and finetuning on OBJECT) that works well for a challenging new task may seem obvious in hindsight, but we respectfully argue that this simplicity is a strength and a technical contribution of this work. As per the reviewer’s suggestion, the comparison of model weights presented next, also confirms the value of initializing with Zero123 as compared to finetuning from Stable Diffusion. Finally, it is not obvious how to construct a training dataset that would allow a Zero123-like model to edit real scenes. As the first work to do so, we consider this a significant and novel contribution. We leave more thorough exploration of model design for future research.
>
> **Comparison of initialization**
>
> This is a great suggestion! We provide a study of different initialization schemes comparing model weight initialization using: (i) Stable Diffusion which was trained for text-to-image; (ii) Zero-123 trained on image-to-image novel-view synthesis on Objaverse (current initialization scheme); (iii) Zero-123 trained on a larger Objaverse-XL [1].
>
> In the table below, we find both Zero-123 based initializations to outperform Stable Diffusion demonstrating the benefits of Zero-123’s novel-view synthesis training by helping the model develop a better implicit understanding of 3D geometry of objects. Initializing with Zero-123 trained on the larger Objaverse-XL dataset achieves similar performance as Zero-123 because Objaverse is already a massive scale pretraining dataset and further scaling Zero-123 style novel-view synthesis pretraining has diminishing returns if any. Scaling limits of our 3D-aware editing training are yet to be explored.
>
> |                   | Seen Objects |       |       |       | Unseen Objects |        |       |       |
> |-------------------|--------------|-------|-------|-------|----------------|--------|-------|-------|
> | Model             | PSNR         | SSIM  | LPIP  | FID   | PSNR           | SSIM   | LPIP  | FID   |
> | Task: Translation |              |       |       |       |                |        |       |       |
> | SD                | 14.373       | 0.264 | 0.51  | 0.101 | 14.351         | 0.253  | 0.51  | 0.102 |
> | Zero123           | 15.21        | 0.3   | 0.472 | 0.244 | 15.2           | 0.292  | 0.477 | 0.253 |
> | XL                | 15.121       | 0.294 | 0.477 | 0.252 | 15.052         | 0.286  | 0.478 | 0.239 |
> | Task: Rotation    |              |       |       |       |                |        |       |       |
> | SD                | 15.074       | 0.368 | 0.43  | 0.089 | 14.558         | 0.359  | 0.438 | 0.095 |
> | Zero123           | 16.859       | 0.382 | 0.429 | 0.248 | 16.279         | 0.366  | 0.447 | 0.236 |
> | XL                | 15.433       | 0.381 | 0.42  | 0.241 | 15.008         | 0.3783 | 0.429 | 0.243 |
> | Task: Insertion   |              |       |       |       |                |        |       |       |
> | SD                | 13.22        | 0.253 | 0.57  | 0.108 | 13.131         | 0.255  | 0.572 | 0.1   |
> | Zero123           | 13.63        | 0.263 | 0.551 | 0.222 | 13.088         | 0.261  | 0.568 | 0.214 |
> | XL                | 13.481       | 0.264 | 0.557 | 0.274 | 13.094         | 0.259  | 0.566 | 0.258 |
> | Task: Removal     |              |       |       |       |                |        |       |       |
> | SD                | 23.882       | 0.576 | 0.263 | 0.117 | 23.352         | 0.542  | 0.27  | 0.115 |
> | Zero123           | 24.98        | 0.585 | 0.249 | 0.236 | 24.661         | 0.568  | 0.26  | 0.24  |
> | XL                | 24.775       | 0.585 | 0.255 | 0.247 | 24.83          | 0.568  | 0.253 | 0.215 |
>
>
> **Related work**
>
> Thank you for these recommendations! We already cited ControlNet, but will add the other three suggestions and adequately discuss them in the related work.  At a high-level, a key difference between these works and ours is that our work focuses not on changing the style, texture or attributes of objects in the scene but rather on enabling 3D-like object rearrangement capabilities via 2D image editing.
>
>
> **Changes in edited image**
>
> Since our method is based on Zero123 and SD, we inherit some of the flaws of these approaches, like creating global changes while targeting local ones.
>
> **Specifying object location**
>
> They are provided in pixel space. Our reason for doing this is to support an intuitive interaction interface (e.g. on a phone) for editing images where a user can simply click on the region of the image they want to be edited. That being said, our modeling framework can easily be extended to incorporate camera coordinates or spatial maps as well if available.
>
> References
> [1] Objaverse-XL: A Universe of 10M+ 3D Objects

---

> > ### Comment · Reviewer_Rpi6 · 2023-08-18
> > **Response to rebuttal**
> >
> > The authors do a great job of addressing most of the pressing concerns. I have additionally gone through the other reviews and agree that the provided dataset has some merits.
> > However, I am still not entirely convinced about the technical novelty of the approach.
> > Although performance trends w.r.t to SD initialization vs Zero-123 initialization are insightful and help highlight the need for the additional viewpoint based finetuning step, it is unclear if the performance difference for these task come from the viewpoint awareness injected by zero-123 or from more data that it sees during finetuning. The edits themselves although not multiview consistent provide a method to generate useful paired data. Additionally, as mentioned above the proposed dataset (if released) would make for a good benchmark.
> > To that end, I would keep the current score and encourages the authors to release the dataset.

---

> > > ### Author Response · Authors · 2023-08-20
> > >
> > > Thank you very much for taking the time to review our work and for giving thoughtful comments in response to our rebuttal. One of the primary goals of this paper is to establish a benchmark for this important task. We will definitely be releasing our code, models and dataset publicly.
> > >
> > > We would also like to clarify that with both models being finetuned on the same data and task, the only difference is in their initialization. Therefore, the difference in performance must come from viewpoint awareness learned by zero123.

---

### Official Review · Reviewer_1T99 · 2023-07-13

**Soundness:** 3 good
**Presentation:** 4 excellent
**Contribution:** 3 good
**Rating:** 5
**Confidence:** 4

**Summary:**

The 3DIT model is a language-guided 3D-aware image editing tool that allows for effective object editing while considering scale, viewpoint, lighting, and object occlusions. The model builds upon previous work in scene rearrangement and image generation, and incorporates a diffusion process to render object transformations. The authors conducted human preference evaluations to measure geometric and lighting consistency, and found that 3DIT outperformed relevant baselines in both categories.

One of the key strengths of 3DIT is its ability to add, remove, or edit shadows to maintain consistency with scene lighting. This is achieved through a shadow generation module that takes into account the position and orientation of the light source, as well as the geometry of the objects in the scene. Additionally, 3DIT accounts for object occlusions by using a novel occlusion-aware rendering module that predicts the visibility of each object in the scene.

The authors also introduced a new benchmark dataset called OBJECT, which consists of 3D scenes with multiple objects and associated natural language descriptions. They trained 3DIT on this dataset and found that it generalized well to images in the CLEVR dataset as well as the real world. This demonstrates the robustness and versatility of the model, and suggests that it could be applied to a wide range of real-world scenarios. Overall, by enabling users to edit objects in a natural and intuitive way, 3DIT opens up new possibilities for creative expression and visual communication.


**Strengths:**

- The 3DIT model is a novel approach to language-guided 3D-aware image editing that builds upon previous work in scene rearrangement and image generation. The model incorporates a diffusion process to render object transformations and uses a novel shadow generation module and occlusion-aware rendering module to maintain consistency with scene lighting and object occlusions.

- The authors conducted human preference evaluations to measure geometric and lighting consistency, and found that 3DIT outperformed relevant baselines in both categories. This demonstrates the effectiveness of the model in producing high-quality, visually consistent image edits.

- The authors trained 3DIT on a new benchmark dataset called OBJECT and found that it generalized well to images in the CLEVR dataset as well as the real world. This suggests that the model is robust and versatile, and could be applied to a wide range of real-world scenarios.

- The potential applications for this technology are vast, including virtual and augmented reality, gaming, and e-commerce. The model could be used to create personalized avatars for virtual reality environments, or to generate realistic product images for e-commerce websites. Additionally, the model could be extended to support more complex scenes and interactions, such as object physics and collision detection.


**Weaknesses:**

- The ablation study is relatively weak. It's unknown which component contributes most to the final performance and which is effective.

- The shadow of the box did not follow the rotation action in the shown GIF. And some artifacts are obvious.

- How did the method choose which box to be moved? Or did it need a handcrafted mask as the selection?

- What is the number of samples used for calculating FID? Normally FID is not reliable of the number of samples is small.

- The comparing table did not include the previous method. I believe several important baselines [a] methods are missing.

References:
[a] Editable free-viewpoint video using a layered neural representation

**Questions:**

N/A

**Limitations:**

See above

---

> ### Author Rebuttal · Authors · 2023-08-10
>
> We thank the reviewer for taking the time to read our submission and for providing valuable feedback. We will now respond to the highlighted questions and concerns.
>
> **Lack of ablation**
> We have extensively evaluated our model's capability using multiple metrics across tasks, single/multitask models, and seen/novel objects. In addition, we now include an ablation of model initialization. To test the effect of the pre-trained initialization, we train two additional versions of the multitask model with different initializations. One is initialized with image-conditioned StableDiffusion [1] and the other with Zero123-XL [2].
>
> In the table below, we find both Zero-123 based initializations to outperform Stable Diffusion demonstrating the benefits of Zero-123’s novel-view synthesis training by helping the model develop a better implicit understanding of 3D geometry of objects. Initializing with Zero-123 trained on the larger Objaverse-XL dataset shows diminishing returns with regards to object-level pretraining. However, the scaling limits of our 3D-aware editing training are yet to be explored and would be an interesting topic for future work.
>
> |                   | Seen Objects |       |       |       | Unseen Objects |        |       |       |
> |-------------------|--------------|-------|-------|-------|----------------|--------|-------|-------|
> | Model             | PSNR         | SSIM  | LPIP  | FID   | PSNR           | SSIM   | LPIP  | FID   |
> | Task: Translation |              |       |       |       |                |        |       |       |
> | SD                | 14.373       | 0.264 | 0.51  | 0.101 | 14.351         | 0.253  | 0.51  | 0.102 |
> | Zero123           | 15.21        | 0.3   | 0.472 | 0.244 | 15.2           | 0.292  | 0.477 | 0.253 |
> | XL                | 15.121       | 0.294 | 0.477 | 0.252 | 15.052         | 0.286  | 0.478 | 0.239 |
> | Task: Rotation    |              |       |       |       |                |        |       |       |
> | SD                | 15.074       | 0.368 | 0.43  | 0.089 | 14.558         | 0.359  | 0.438 | 0.095 |
> | Zero123           | 16.859       | 0.382 | 0.429 | 0.248 | 16.279         | 0.366  | 0.447 | 0.236 |
> | XL                | 15.433       | 0.381 | 0.42  | 0.241 | 15.008         | 0.3783 | 0.429 | 0.243 |
> | Task: Insertion   |              |       |       |       |                |        |       |       |
> | SD                | 13.22        | 0.253 | 0.57  | 0.108 | 13.131         | 0.255  | 0.572 | 0.1   |
> | Zero123           | 13.63        | 0.263 | 0.551 | 0.222 | 13.088         | 0.261  | 0.568 | 0.214 |
> | XL                | 13.481       | 0.264 | 0.557 | 0.274 | 13.094         | 0.259  | 0.566 | 0.258 |
> | Task: Removal     |              |       |       |       |                |        |       |       |
> | SD                | 23.882       | 0.576 | 0.263 | 0.117 | 23.352         | 0.542  | 0.27  | 0.115 |
> | Zero123           | 24.98        | 0.585 | 0.249 | 0.236 | 24.661         | 0.568  | 0.26  | 0.24  |
> | XL                | 24.775       | 0.585 | 0.255 | 0.247 | 24.83          | 0.568  | 0.253 | 0.215 |
>
> **Inconsistent shadow in GIF**
> As mentioned in the common statement, the original version of the OBJECT dataset used a single directional light source which was very different than lighting found in CLEVR (the dataset used for making the GIFs) and real images. Hence, learning to model lighting and shadows on the OBJECT dataset had limited generalization to out-of-distribution images found in CLEVR. In-line with reviewer suggestions, we improved the dataset with a more realistic 3-point lighting (see the Figures B for examples of the improved dataset) that reduced the sim-to-real gap as well as the domain gap to CLEVR. As can be seen in figure H, the new shadow of the blue cube more accurately moves in response to the rotation.
>
> **Artifacts in the GIF**
> Like other SOTA image editing diffusion models, the model often makes small changes outside the region of interest. These artifacts are often difficult to notice in isolated edited images but become clear when making a sequence of edits and compiling them into a video or GIF. Techniques for improving temporal consistency in video-editing literature might be applicable, but our current benchmark and method focuses on single atomic edits as an important first step.
>
> **How did the method choose which box to be moved? **
> We do not need to provide a mask for the box because our method is conditioned on a language description of the object. We identify the box with the prompt “a blue box” for the rotation GIF. Please see the supplementary material of our submission for the rest of the prompts in the GIFs.
>
> **Number of samples for FID**
> We calculated FID with 1024 samples.
>
> **Additional baseline**
>
> In this work, we proposed a novel problem of 3D-aware image editing. While there aren't existing methods that solve this exact problem, we do compare to relevant baselines that involve chaining off-the-shelf models together. The suggested baseline, from the paper “Editable free-viewpoint video using a layered neural representation”, is interesting, but requires video from 16 different cameras. Our intention is to allow users to edit photos from a single RGB image, making this baseline out of the scope of our problem formulation. However, we will add this paper to the discussion in the related work.
>
> References
> [1] High-Resolution Image Synthesis with Latent Diffusion Models
> [2] Objaverse-XL: A Universe of 10M+ 3D Objects

---

### Author Rebuttal · Authors · 2023-08-10

# Common statement

We are encouraged by all the positive comments and thank all of the reviewers for their valuable feedback. Reviewers found our model to be “a novel approach to language guided 3D-aware image editing” (Reviewer 1T99), “the manipulations possible with the method can preserve the 3D properties of the scene” (Reviewer yCLA), and that our dataset is “guaranteed to be 3D-correct, and would be useful for research” (Reviewer SoaB), among other positive comments which we are grateful for.  In this section, we will address the questions and concerns that were shared in common by multiple reviewers.

**Reviewers eZ2Z and SoaB expressed concerns that the real-world performance of our model may be limited due to the realism of the dataset.**

First, we note that in spite of the less realistic training data, we were surprised by the generalization of our models to real images. This is a significant finding that establishes training on simulated 3D-aware image editing examples as a promising direction for this challenging object-centric editing task.

Second, in line with reviewers suggestions, we have improved the realism of our dataset in 3 ways: (i) As per common practices in professional film and photography, we implemented a 3-point lighting system that automatically aligns itself with the viewpoint of the camera, thereby shading objects in a way that better reveals their true 3D form; (ii) We added real-world environmental lighting into our dataset with 360-degree HDRI captures from both indoor and outdoor scenes under a variety of lighting conditions. Not only does this give scenes in our dataset realistic backgrounds, but the light emanated from these captures is integrated into the ray-tracing process during rendering, so that all aspects of the scene benefit from a more realistic lighting distribution; and (iii) We replaced the ground textures in our dataset with more realistic ones that have normal, roughness and displacement maps. Please see Figures A, B, and D in the attached pdf for examples from the new dataset.

Third, we retrained our models on this more realistic dataset and find that the in-domain performance follows similar trends as reported in the paper, but the models generalize better to out-of-domain data like CLEVR and the real world images. For results, please see a table comparing human evaluation on the old and new model in response to Reviewer yCLA. Please also see figures A, G, H and I for qualitative results. We will update the draft with all recomputed metrics for our new models on this improved dataset. For brevity, we report a table with PSNR metrics which can be found in the response to Reviewer 1T99.


**Reviewers 1T99, Rpi6, yCLA, eZ2Z and SoaB raised concerns about the limitations of our method.** The reviewers correctly identified some of the failure modes of our approach which we address below:
 1. **Unintended global changes / artifacts in GIF:** Like other SOTA diffusion based editing / inpainting models, our model may introduce minor artifacts outside the region of interest specially for out-of-distribution images such as CLEVR and real images. These artifacts are often difficult to notice in isolated edited images but become clear when making a sequence of edits and compiling them into a video or GIF as pointed out by Reviewers 1T99, yCLA, Rpi6, and eZ2Z. Techniques for improving temporal consistency in video-editing literature might be applicable, but our current benchmark and method focuses on single atomic edits as an important first step.

 2. **Inaccurate lighting and shadows:** The original dataset at the time of submission used a single directional light source leading to less realistic lighting in our training dataset. With the improvements to the training data, the scenes are now lit in a much more realistic manner leading to noticeably better shadow rendering behavior in our models. Please see the table in the response section of Reviewer yCLA for a human evaluated comparison between the new and old models. We show frames of the GIFs from our old and new model for comparison in Figure I.

However, we emphasize that our main contribution is to introduce the challenging and novel task of object-centric 3D-aware image editing and to create a benchmark for training and evaluation. Our benchmark and model will be useful for future research in this area.

## Additional experiment requests

**Initialization Ablation.** In response to Reviewers 1T99 and Rpi6’s suggestions for model ablations, we provide a study of different initialization schemes comparing model weight initialization using: (i) Stable Diffusion which was trained for text-to-image; (ii) Zero-123 trained on image-to-image novel-view synthesis on Objaverse (current initialization scheme); (iii) Zero-123 trained on a larger Objaverse-XL [1].

In the table listed in the response to Reviewer 1T99, we find both Zero-123 based initializations to outperform Stable Diffusion demonstrating the benefits of Zero-123’s novel-view synthesis training by helping the model develop a better implicit understanding of 3D geometry of objects. Initializing with Zero-123 trained on the larger Objaverse-XL dataset achieves similar performance as Zero-123 because Objaverse is already a massive scale pretraining dataset and further scaling Zero-123 style novel-view synthesis pretraining has diminishing returns if any. Scaling limits of our 3D-aware editing training are yet to be explored.

**Human-Evaluation on Real Images.** We also provide quantitative human-evaluation on real images as requested by Reviewers yCLA and SoaB. We find that human evaluators overwhelmingly prefer results from our method over baselines. Among all tasks, insertion is the most challenging for our model. Please see the tables in the section of Reviewer yCLA.

References
[1] Objaverse-XL: A Universe of 10M+ 3D Objects

---

### Decision · Program_Chairs · 2023-09-21

**Decision:**

Accept (poster)

**Comment:**

Evenly the scores are not that high, all reviewers have consistent opinion and stand on the positive side. Although the performance is limited, all reviewers and the AC agree on the value of the proposed dataset.